# ^67^Cu Production Capabilities: A Mini Review

**DOI:** 10.3390/molecules27051501

**Published:** 2022-02-23

**Authors:** Liliana Mou, Petra Martini, Gaia Pupillo, Izabela Cieszykowska, Cathy S. Cutler, Renata Mikołajczak

**Affiliations:** 1Legnaro National Laboratories, National Institute for Nuclear Physics, Legnaro, 35020 Padova, Italy; liliana.mou@lnl.infn.it (L.M.); gaia.pupillo@lnl.infn.it (G.P.); 2Department of Environmental and Prevention Sciences, University of Ferrara, 44121 Ferrara, Italy; petra.martini@unife.it or; 3National Centre for Nuclear Research, Radioisotope Centre POLATOM, 05-400 Otwock, Poland; izabela.cieszykowska@polatom.pl; 4Brookhaven National Laboratory, Collider Accelerator Department, Upton, NY 11973, USA; ccutler@bnl.gov

**Keywords:** Cu-67, copper radionuclides production, radiopharmaceuticals, theranostics

## Abstract

Is the ^67^Cu production worldwide feasible for expanding preclinical and clinical studies? How can we face the ingrowing demands of this emerging and promising theranostic radionuclide for personalized therapies? This review looks at the different production routes, including the accelerator- and reactor-based ones, providing a comprehensive overview of the actual ^67^Cu supply, with brief insight into its use in non-clinical and clinical studies. In addition to the most often explored nuclear reactions, this work focuses on the ^67^Cu separation and purification techniques, as well as the target material recovery procedures that are mandatory for the economic sustainability of the production cycle. The quality aspects, such as radiochemical, chemical, and radionuclidic purity, with particular attention to the coproduction of the counterpart ^64^Cu, are also taken into account, with detailed comparisons among the different production routes. Future possibilities related to new infrastructures are included in this work, as well as new developments on the radiopharmaceuticals aspects.

## 1. Introduction

Copper-67 (^67^Cu) (t_1/2_ = 2.58 d), the longest-living radioisotope of Cu, is of paramount importance because of its simultaneous emissions of β^−^ radiation (mean β^−^ energy: 141 keV; E_β^−^_max: 562 keV), useful for therapeutic treatments and γ-rays (93 and 185 keV), suitable for single-photon emission computed tomography (SPECT) imaging. In fact, the ^67^Cu mean β^−^-emission energy of 141 keV (Eβ−max: 562 keV) is slightly higher than that of Lutetium-177 (^177^Lu, β^−^-emission energy of 133.6 keV, Eβ−max: 497 keV). ^67^Cu decay characteristics make it one of the most promising theranostic radionuclides and its long half-life makes it suitable for imaging in vivo slow pharmacokinetics, such as monoclonal antibodies (MoAbs) or large molecules [1]. ^67^Cu, studied for decades for radioimmunotherapy [2,3,4], is currently under the spotlight in the international community, as highlighted by the recent IAEA Coordinated Research Project (CRP) on “Therapeutic Radiopharmaceuticals Labelled with New Emerging Radionuclides (^67^Cu, ^186^Re, ^47^Sc)” (IAEA CRP no. F22053) [5,6]. ^67^Cu can also be paired with the β^+^ emitters ^64^Cu, ^61^Cu, and ^60^Cu to perform pretherapy biodistribution determinations and dosimetry using positron emission tomography (PET) systems. Table 1 presents the decay characteristics of ^67^Cu and ^64/61/60^Cu-radionuclides, as extracted from the NuDat 3.0 database [7].

Among copper radionuclides, only ^64^Cu has been widely used for preclinical and clinical PET studies due to its moderate half-life (t_1/2_ = 12.7 h), low positron energy, and availability [8]. Given that copper radioisotopes are chemically identical, the same bifunctional chelators that have been developed for ^64^Cu radiopharmaceuticals can be used directly for ^67^Cu (and ^61/60^Cu) labeling. While the production techniques for ^64^Cu are well known and are usually based on the ^64^Ni(p,n)^64^Cu, and ^64^Ni(d,2n)^64^Cu reactions [9], the use of ^67^Cu has been prevented by a lack of regular availability of sufficient quantities for preclinical and clinical studies. Only recently has ^67^Cu become available in the U.S. through the Department of Energy Isotope Program (DOE-IP), in quantities and purities that are sufficient for medical research applications [10]. The investigation of ^67^Cu supply worldwide is, therefore, a crucial point, and this review presents the state-of-art in ^67^Cu production and medical applications.

## 2. Production Methods of ^67^Cu

### 2.1. Accelerator-Based Production

#### 2.1.1. Charged-Particle Induced Reactions

While ^64/61/60^Cu radionuclides can be produced via low energy medical cyclotrons [11], for ^67^Cu production, intermediate proton energies are needed, as shown by the well-known cross sections on zinc targets. The main route is the ^68^Zn(p,2p)^67^Cu reaction, studied for decades, whose excitation function has been recommended by the International Atomic Energy Agency (IAEA) [12]. This production method is feasible at intermediate proton-beams (E < 100 MeV), but it can be exploited also at higher energies [13]. The use of enriched ^68^Zn targets is mandatory to reduce the coproduction of other Cu-radionuclides affecting the radionuclidic purity (RNP) of the final product [14]. However, the coproduction of ^64^Cu cannot be avoided, as also indicated by the ^68^Zn(p,x)^64^Cu cross section reported by the IAEA up to 100 MeV. 

At low-energy proton beams, the ^70^Zn(p,α)^67^Cu reaction is feasible up to 30 MeV, without the coproduction of ^64^Cu [15]. Due to the low value of the cross section (the maximum at 15 MeV is ca. 15 mb), this production route provides quite a low yield, i.e., 5.76 MBq/µAh for the 30–10 MeV energy range (corresponding to a 1.84 mm thick target). On the other hand, when using ^70^Zn targets and higher proton energies, it is possible to reach a higher ^67^Cu yield, though with the coproduction of ^64^Cu [16,17]. 

Considering gallium as target material, it is possible to investigate the use of ^71^Ga, whose natural abundance is 39.892% [7]. The cross section of the ^71^Ga(p,x)^67^Cu nuclear reaction is very low, about 2.6 mb at 33 MeV, and so far few literature data are covering the 20–60 MeV energy range [18]; however, the low cross section makes this route of ^67^Cu production not feasible. 

In order to estimate the influence of various parameters in the proton-based production of ^67^Cu, we used the IAEA tool ISOTOPIA [19] and the recommended nuclear cross sections for calculations; when the IAEA cross sections values were not available, a fit of the literature data were used [20]. The ^67^Cu activity was calculated for 62 h irradiation, equivalent to a ^67^Cu Saturation Factor SF = 50% and a ^64^Cu SF = 97%. Results are shown in Table 2, calculated at the end of bombardment (EOB) and 24 h after the EOB [13]. Table 2 also presents the ^67^Cu yield when 70 MeV protons are exploited on a multi-layer target composed of enriched ^68^Zn and ^70^Zn layers, a configuration that optimizes ^67^Cu production and minimizes ^64^Cu coproduction [21].

Deuteron beams can also be exploited for ^67^Cu production. As in the case of proton beams, the use of enriched targets is mandatory to limit the coproduction of contaminant Cu-radionuclides. In addition to this, when using ^nat^Zn targets, the low production cross section [20] also leads to a low ^67^Cu yield. The reported data on the ^70^Zn(d,x)^67^Cu cross section include a measurement from 2012 [22] and a recent work of 2021, describing an experimental campaign up to 29 MeV [23]. The use of ^70^Zn targets seems promising in the energy window 26–16 MeV, giving a ^67^Cu yield of 6.4 MBq/µAh [23]. In order to use higher deuteron beam energies it is important to measure the ^70^Zn(d,x)^64^Cu cross section (^64^Cu threshold energy is E_THR_ = 26.5 MeV [7]), whose values are not reported in the literature [20].

Among the charged-particle induced reactions, the α-beams have also been explored. The main ^67^Cu production route relies on the use of enriched ^64^Ni targets (natural abundance 0.9255%). The enrichment level of the target material is again a key parameter, because of the coproduction of ^64^Cu through the ^nat^Ni(α,x)^64^Cu reaction. Recently published new data of the ^64^Ni(α,p)^67^Cu reaction up to 50 MeV [24] are in agreement with previous results [25,26,27,28]. This latest work also reports the ^67^Cu yield, after 24 h irradiation and 30 µA of beam current. 

To compare the α-based production route with proton- and deuteron-induced nuclear reactions, Table 3 presents the calculated ^67^Cu (and ^64^Cu) production yields, assuming 100% enriched targets and the same irradiation conditions (^67^Cu SF = 24%, ^64^Cu SF = 73%). The ^61^Cu and ^60^Cu contaminants are not included in the calculations, since their half-lives are significantly shorter than ^67^Cu half-life; thus, their impact on the RNP is relevant only soon after the EOB.

The calculations reported in Table 3 show that the most convenient route to obtain pure ^67^Cu (without ^64^Cu coproduction) is by using deuteron beams and ^70^Zn targets. Moreover, the use of α-beams and ^64^Ni targets provides pure ^67^Cu. Intense linear accelerators (ca. mA current) for α-particles, requiring specific targets able to withstand such high currents that are still to be designed, are soon foreseen but not yet available. The proton-induced reactions with a ^68^Zn target and a ^68/70^Zn multi-layer target configuration seem to be a promising option if some ^64^Cu coproduction is acceptable, since these routes provide a larger ^67^Cu yield in comparison with the ^70^Zn(d,x)^67^Cu route. In all cases, by changing the projectile type and/or its energy and/or the target material, it is possible to adapt the ^67^Cu production yield and the profile of contaminants.

#### 2.1.2. Photonuclear Production

The photonuclear production for ^67^Cu using bremsstrahlung photons with an e-LINAC accelerator has been studied for decades [29,30,31,32,33,34,35,36]. Recently, a large enriched ^68^Zn target (55.5 g) was irradiated with 40 MeV e-LINAC for 53.5 h, obtaining 62.9 GBq (1.7 Ci) without detecting ^64^Cu [10]. The threshold energy of the ^68^Zn(γ,x)^64^Cu reaction is E_THR_ = 27.6 MeV [37]. For the photonuclear reaction, the use of enriched target material is mandatory to avoid the coproduction of the following radionuclide impurities: ^63^Zn, ^65^Zn, and ^64^Cu. According to the DOE Isotope Program, ^67^Cu is being routinely produced by the Argonne National Laboratory via a photonuclear reaction at its low-energy accelerator facility (LEAF). Approximately 1 curie per batch can be provided, with radionuclidic purity of >99% and a specific activity >1850 GBq/mg (>50 Ci/mg) ^67^Cu/total Cu at EOB [38]. Commercial entities have indicated they are pursuing this route of production, although no commercial suppliers are online as of the date of this publication. 

### 2.2. Reactor-Based Production

^67^Cu can be produced in a reactor via the ^67^Zn(n,p)^67^Cu nuclear reaction with fast neutrons and, after the separation process, it is possible to obtain ^67^Cu in a n.c.a. form. The enrichment of the ^67^Zn target material is crucial to obtain quantities of ^67^Cu suitable for medical application. In natural zinc, the abundance of ^67^Zn is only 4.04%, while ^64^Zn contributes to 49.17% [7] and the neutron cross-section for the ^64^Zn(n,p)^64^Cu nuclear reaction is much higher, hence the amount of ^64^Cu produced is an order of magnitude higher than that of ^67^Cu. The use of enriched ^67^Zn not only limits the contribution of ^64^Cu but also increases the irradiation yield of ^67^Cu. However, due to the low cross section value, the ^67^Zn(n,p)^67^Cu nuclear reaction requires a high flux of fast neutrons exceeding 10^14^ n cm^−2^s^−1^ [3,39,40,41]. In the fast neutron flux of 4.4 × 10^14^ n cm^−2^s^−1^ (En > 1 MeV) the saturation yield of ^67^Cu at the EOB was 4.14 ± 0.37 GBq/mg of ^67^Zn, these values were dependent on the neutron flux and the position of the target in the reactor [37,40]. When the thermal and the fast neutron fluxes were 1.3 × 10^12^ n cm^−2^s^−1^ and 1.5 × 10^12^ n cm^−2^s^−1^, respectively, only 630 kBq/mg Zn of ^67^Cu was produced after a 5 h irradiation of target containing around 50 mg of 93.4% enriched ^67^ZnO. In the obtained mixture of ^64^Cu and ^67^Cu, the latter contributed to less than 30% of the total radioactivity. In order to increase the amount of ^67^Cu produced by this reaction, irradiation in a nuclear reactor with higher fast neutron flux, and for longer periods of irradiation are required [42]. More detailed summation of reported neutron irradiation yields for the ^67^Zn(n,p)^67^Cu nuclear reaction has been previously reported [43,44]. 

In nuclear reactors with a high ratio of thermal to fast neutrons, the coproduction of ^65^Zn (t_1/2_ = 243.93 d) is unavoidable because of the presence of ^64^Zn, particularly in the natural zinc target material, and the relatively high cross section of ^64^Zn(n,γ)^65^Zn nuclear reaction with thermal neutrons. Although ^65^Zn can be separated from ^67^Cu during the target processing, due to its long half-life, ^65^Zn contaminates the recycled target material. Thermal neutron shielding made of materials with high neutron capture cross sections, such as boron, cadmium or hafnium, may reduce this contamination [45,46,47]. A boron nitride shield of 3.48 g/cm^3^ density and around 4 mm thickness reduced the thermal flux in the sample holder from about 10^13^ n/cm^2^/s to approximately 10^10^ n/cm^2^/s, resulting in a production of about 11 times higher for ^67^Cu than ^65^Zn and in reducing the ^65^Zn production in ^67^Zn targets by a factor of 66 [45]. Other reported by-products included ^58^Co produced via the ^58^Ni(n,p)^58^Co reaction, ^67^Ga, and interestingly, ^182^Ta, which can be produced via thermal neutron capture on ^181^Ta present in the target material [45]. Potentially, all these contaminants can be removed in the chemical separation of ^67^Cu. 

The measurements of the ^71^Ga(n,n + α)^67^Cu nuclear reaction show an increasing trend from 13 MeV to 20 MeV, reaching a maximum value of ca. 20 mb [20]. Because of these quite low cross section values, ^71^Ga targets are impracticable for ^67^Cu production.

### 2.3. Targetry

In the charged-particle induced nuclear reactions, enriched Zn is the most commonly used target material, although Ni has also been used with α-beams. The accelerator-based production requires the use of highly enriched target material to achieve high radionuclidic and chemical purity of the product thus making the target very expensive and necessitating target recovery and recycling to minimize the production costs. Targets composed by a set of thin foils in the well-know “stacked-foils” configuration have been used in preliminary studies for nuclear cross section measurements, with enriched Zn foils produced by electrodeposition [48] or by lamination from Zn metal powder [16]. In the accelerator production of ^67^Cu, mainly thick solid targets have been used, either in the form of metallic foil/coin or in the oxide form [13], despite the low melting point of Zn. 

A multilayer target configuration, based on the use of both ^68^Zn and ^70^Zn enriched materials, has been recently patented for ^67^Cu proton induced production [21]. This target configuration, shown in Figure 1, is beneficial with an increase of 48% in the ^67^Cu production and a 12% decrease of ^64^Cu coproduction, with respect to a thick monolayer of ^68^Zn in the energy range 70–35 MeV and 24 h irradiation (Table 3). In addition to this, in the low energy range (E < 30 MeV), it is possible to add another ^70^Zn layer to exploit the (p,α) reaction to increase the ^67^Cu yield. It is important to underline that, in this final ^70^Zn layer (covering the low-energy region), there is no coproduction of ^64^Cu. In the patent, it is suggested to apply a radiochemical process to each target layer individually, in order to recover each enriched material separately (^68^Zn and ^70^Zn). Moreover, the user can decide to combine the final solutions (containing the ^67^Cu/^64^Cu mix of radionuclides and the pure ^67^Cu) or to use them separately to label the desired radiopharmaceuticals. 

Electroplating of enriched Zn or Ni on a gold, titanium, aluminum, or gold-plated copper backing is the most popular target fabrication technique. With this method, the target thickness can be adapted to the optimal specific beam energy range and thicknesses up to 80 mm have been reported [6,13,49,50]. For the production of ^67^Cu via the ^64^Ni(α,p)^67^Cu nuclear reaction, Ohya et al., used enriched ^64^Ni target material (^64^Ni 99.07%) [50]. Sublimation and the casting process have also been used to produce massive zinc ingot targets (50–100 g) for photonuclear production [10,31]. 

Zinc oxide targets have been used for both photonuclear- and nuclear reactor-based production of ^67^Cu. Recently, ZnO was pressed and wrapped in aluminum foil and used as a target for the photonuclear production [24]. In the targets for neutron irradiation in nuclear reactors, zinc oxide powder was encapsulated in a quartz ampule and aluminum cans or sealed in a polyethylene bag and sandwiched between two Ni foils [6,51,52,53,54].

^nat^ZnO nanoparticles were compared to ^nat^ZnO powder in the target irradiation at a fast neutron flux. Both targets of the same mass, 1.0 g each, were irradiated for 30 min in the fast neutron flux of 1.4 × 10^13^ n·cm^−2^s^−1^, showing an increase in the ^67^Cu activity produced that almost doubled (0.0168 MBq vs. 0.0326 MBq) when ^nat^ZnO nanoparticles target was used [51]. 

### 2.4. Radiochemistry

Strictly related to the target configuration is radiochemical target processing, aimed at transforming the produced ^67^Cu in the solution suitable for radiolabeling. Since the commercially available target materials, even with the highest achievable enrichment of the desired isotope, still contain elemental and isotopic contaminants, their irradiation leads to collateral production of several zinc, cobalt, and gallium radionuclides. The method for the separation of Cu from the dissolved target material must therefore ensure not only the removal of the bulk target material but also of these specific side products. The process should also aim to recover the enriched target material, which, in general, is very expensive. The separation process can be accomplished in one step or in a combination of more steps using conventional separation methods, such as ion chromatography, solvent extraction, precipitation, sublimation, etc., to isolate the desired radionuclide and eliminate contaminants.

Solvent extraction with dithizone (diphenylthiocarbazone) dissolved in a water immiscible medium, e.g., CCl_4_, CHCl_3_, was proposed because of its selectivity for ions of various metals depending on the pH. Dithizone is selective for Cu in the pH range 2–5, for Zn in the pH range 6.5–9.5, and for Ni in the pH range 6–9. Using this method ^67^Cu could be separated with dithizone even from large amounts of Zn, up to 5 g, and from other co-produced impurities [36,53,55,56]. The Cu is then back extracted into an aqueous phase by shaking the organic solution with 7 M HCl mixed with H_2_O_2_, resulting in dissociation of Cu-chelate [36,53,56]. This technique is proposed as a first step for Cu separation from bulk target material, which can be then followed by further purification using chromatographic methods [57,58]. Recently, the dithizone-based solvent extraction and back extraction of Cu from large excess of Zn have been implemented in a microfluidic system [59].

A new method combining the solvent extraction and anion exchange separation techniques into a single separation was proposed by Dolley et al., by using an AmberChrom CG-71 dithizone-impregnated resin. At first, the coproduced ^66,67^Ga were removed with an untreated AmberChrom CG-71 resin. Next, ^67^Cu was separated from ^65^Zn and ^56,57,58^Co on the dithizone-based solid phase extraction chromatographic column, which retained Cu radioisotopes [60]. Improvements of this method were lately proposed for the separation of Cu from large amounts of zinc by using a single modified dithizone (diphenylthiocarbazone) Amberlite^®^ XAD-8 (20–60 mesh) chelating resin [61]. Dithizone is sensitive to oxidation, forming diphenylthiocarbodiazone when exposed to light and heat. Thus, dithizone impregnated resin must be relatively freshly prepared before use. A semi-automated separation module adjusted to operate in a shielded facility employing liquid–liquid extraction of Cu from Zn target was developed [55]. 

Another extractant proposed for the separation of ^67^Cu from ^67^ZnO irradiated in nuclear reactor was thenoyltrifluoroacetone (TTA) in benzene [44]. Despite the high separation yield, the content of organic extractant residue in the final product is a drawback of this extraction methods. In the past, other chelating agents for Cu were proposed, such as cupferron and diethyldithiocarbamate [57]. 

Electrolysis has also been used for the separation of Cu from Zn and Ni target material [39,56,62]. Although the purity of the final Cu solution isolated by conventional electrolysis was adequate for antibody labeling, the process was very time consuming and ^67^Cu losses occurred at each electrolytic step. Thus, this approach was suggested to be unsuitable for routine production.

While most researchers have reported on the application of electrolytic separation under an external electromotive force (EMF), spontaneous electrochemical separation of Cu from proton- or neutron- irradiated zinc targets has been investigated [39]. The process is simple and can be easily automated and adopted to work in hot cells, which is advantageous over conventional electrolysis. Moreover, without an externally applied voltage, the process is more selective in separating Cu from interfering metal ions, such as Fe, Co, or Ni, due to the elimination of hydrogen overvoltage and the preservation of the Pt electrode. Time of the process was only 30 min. As result of the spontaneous electrodeposition in the processing of the proton-irradiated ZnO target, the separation factors of ^67^Cu from isotopes of Co, Cr, Fe, Ga, Mn, Ni, and V has been reported [39]. Separation factors of >1 × 10^7^ from grams of Zn can be achieved using this method, obtaining highly pure n.c.a. ^67^Cu. The overall separation factors ranged from 7 × 10^3^ for ^57^Ni to 9 × 10^4^ for ^58^Co, and the separation factor from ^67^Ga was >1.3 × 10^4^. 

Ion exchange is frequently used in the separation of radionuclides for medical application due to its high efficiency, reproducibility, and ease in automation. Though, the separation of Cu from Zn and contaminants could be efficiently accomplished (recovery yield 92–95%) using three ion exchange columns (cation exchange resin AG50W-X8, anion exchange resin AG1-X8 and chelating resin Chelex 100). The method is laborious and includes several time consuming evaporation steps [13,63]. An alternative co-precipitation of ^67^Cu from bulk Zn using H_2_S gas with an excess of silver nitrate and consecutive separation of precipitate by filtration was proposed [63]. Compared with ion exchange, this new process of ^67^Cu separation was completed in less than 3 h with similar recovery. 

In the separation of ^67^Cu produced by irradiation of a Ni target, the same procedures that have been used in the routine production of ^64^Cu by the ^64^Ni(p,n)^64^Cu nuclear reaction can be adopted. The Cu/Ni separation and purification is typically a one-step procedure, in which an ion exchange resin (in anionic or cationic form, AG1-X8 and AG50W-x8 respectively) is used for eluting Ni, Co, and Cu with various acid concentration or acid/organic solvent ratio [50,64,65,66]. 

A summary of the procedures that have been investigated for the separation of ^67^Cu from an irradiated zinc target, either in metallic or oxide form, is presented in Table 4.

### 2.5. Recovery

When highly enriched target materials are needed, their cost makes the target recovery and recycling mandatory [6]. The choice of the recovery technique depends on the method used for ^67^Cu separation and the expected chemical form of the recovered material.

When electrodeposition is the procedure selected for target production, the recovery of either Zn or Ni target materials is rather simple and requires only a few chemical steps for re-establishing the electrodeposition conditions, as described by Medvedev et al. [13]. A pre-cleaning of the target material before its electrodeposition can be performed to minimize the impurities in the recycled target and to improve the specific activity of the final product. Before Zn target material recovery, for example, the radiogallium can be removed by using a cation exchange column AG50W-X8 as described by Ohya et al. [63].

In the procedure reported by Shikata in 1964, zinc was recovered in the oxide form by evaporating to dryness the 0.01 M HCl zinc-rich solution eluted from the anion exchange resin, dissolving the obtained zinc chloride in water, heating up the solution to about 70 °C and adding ammonium oxalate to induce precipitation of zinc oxalate. The precipitate was then filtered, washed and ignited to constant weight [42]. 

Similarly, after the isolation of ^67^Cu using solvent extraction with TTA/benzene, zinc oxide target material was recovered after evaporation of the aqueous solution under a stream of argon, followed by dissolution of the obtained solids in 2 N HCl, and finally loading the solution on a Bio-Rad resin column [44]. Elution of Zn^2+^ with 6 N HNO_3_ was then evaporated under argon stream. The resulting dry ZnO was transferred to a furnace and heated at 350 °C for 2 h. 

Another recovery technique was reported by Ehst et al., through the sublimation Cu/Zn separation process by recovering sublimed Zn with negligible losses [31]. 

### 2.6. Quality of ^67^Cu as Radiopharmaceutical Precursor

According to Ph. Eur. monograph (0125) [67], a radionuclide precursor is any radionuclide produced for radiolabeling of another substance prior to administration. 

In the absence of a pharmacopoeia monograph specific for ^67^Cu, the proposed quality control protocol for produced ^67^Cu includes identity, radioactivity, specific activity, radionuclidic purity, radiochemical purity, and chemical purity [5,6,50,63].

#### 2.6.1. Identity

Radionuclide identity is determined by assessing the physical characteristics of the radionuclide emissions. The γ-line at the energy Eγ = 184.6 keV (intensity Iγ = 48.7%) has been used to evaluate the activity of ^67^Cu [31]. In order to check that there is no contamination of ^67^Ga in the product, we suggest verifying that the same activity of ^67^Cu is obtained from both 184.6 keV (48.7%) and the 300.2 keV (0.797%) γ-lines. This additional control is due to the γ-lines emitted by ^67^Ga (Eγ = 184.6 keV, Iγ = 21.41% and the Eγ = 300.2 keV, Iγ = 16.64%) having the same energies of ^67^Cu but with different intensities [7].

#### 2.6.2. Specific Activity

Specific activity (SA) is defined as radioactivity per unit mass of the product [68]. Several factors that could affect the specific activity of the final product include cross-sections, target impurities, secondary nuclear reactions, target burn-up, and post-irradiation processing periods. 

The chosen production route may be optimized to limit the production of stable copper isotopes. However, the contamination with stable copper (^63,65^Cu) coming from the materials and chemicals involved in the production and separation is difficult to control. Special care has to be taken employing metal free chemicals and tools. It is therefore fundamental to determine the quantity of stable copper in the final product through inductively coupled plasma with optical emission spectrometry (ICP-OES) or mass spectrometry (ICP-MS) detection techniques.

#### 2.6.3. Radionuclidic Purity

Radionuclidic purity (RNP) is determined by γ-spectrometry using a HPGe detector calibrated using standard sources, coupled with a multichannel analyzer [50]. The coproduced short-lived ^60,61,62,66,68,69^Cu radioisotopes do not significantly affect the RNP of ^67^Cu [5]. However, the coproduction of the longer-lived ^64^Cu in most production routes is unavoidable and the ^64^Cu/^67^Cu ratio is an important factor defining the quality of ^67^Cu. Ohya et al. [50] determined the activities of ^61^Cu, ^64^Cu, ^67^Cu, and ^65^Zn considering the ^64^Ni(α,p)^67^Cu reaction through the measurement of the following γ-lines, respectively: Eγ = 656.0 (10.8%), 1345.8 (0.473%), 184.6 (48.7%), and 1115.5 (50.60%) keV. Based on the analytical method proposed by Van So et al. [69], the activities of ^67^Cu and ^67^Ga can be evaluated from the measurements of their mixed signal in the γ-lines at 184.6 keV (48.7% for ^67^Cu and 21.2% for ^67^Ga), 209.0 keV (0.115% ^67^Cu and 2.4% ^67^Ga), 300.2 keV (0.797% ^67^Cu and 16.8% ^67^Ga), and 393.5 keV (0.22% ^67^Cu and 4.68% ^67^Ga) [63]. With this method, it is possible to infer the ^67^Cu and ^67^Ga activities without a radiochemical separation process. Similarly, Pupillo et al., proposed the method to correct for the residual Ga-activity in copper-solution [16] and Nigron et al., performed the measurement of the ^70^Zn(d,x)^67^Cu cross section [23]. The presence of the following impurities can be determined in the ^67^Cu-solution considering the main radionuclides γ-lines: ^66^Ga (1039 keV, 37%) ^58^Co (810.8 keV, 99.45%), ^65^Zn (1116.0 keV, 50.04%), ^57^Ni (1377.6 keV, 81.7%), and ^105^Ag (344.5 keV, 41.4%). It is important to remember that the impurity profile depends on the nuclear reaction route and on the specific irradiation conditions, in addition to the radiochemical processing of the target. 

#### 2.6.4. Chemical Purity

The chemical purity of radionuclides for medical applications refers to the determination of metal cations, which may compete with the radiometal in the chelator for complex formation. Impurities, such as Zn^2+^, Fe^3+^, Co^2+^, Ni^2+^ can dramatically influence the radiolabeling yield of common chelators such as DOTA (1,4,7,10-tetraazacyclododecane-1,4,7,10-tetraacetic acid). They can arise from solvents used in target processing or can be eluted from resins used for isolation of radiometal from bulk target solution. Another possible contaminant is the residual organic solvent or other organics remaining in the product after liquid–liquid or solid phase extraction. 

Limits for individual metal impurities for radionuclides intended for radiopharmaceutical preparation are described in Ph. Eur. monographs. For example, in the case of ^177^Lu solution for labeling the limit of Cu is 1.0 μg/GBq, Fe: 0.5 μg/GBq, Pb: 0.5 μg/GBq, and Zn: 1.0 μg/GBq [70]. Such requirements have not been yet established for ^67^Cu. Metallic impurities present in radionuclide solutions are typically determined using ICP-OES or ICP-MS techniques. For evaluation of the chemical purity of ^67^Cu the ICP-OES, ICP-MS, and anodic stripping voltammetry were used for the determination of zinc and copper impurities [5,6]. Commercially available colorimetric test kits for the determination of Zn, Fe, or Ni in the final product were also mentioned [5]. Although this method is less sensitive, it enables sub-ppm analysis and can be used as a preliminary analysis [5]. The macrocyclic copper chelator of 1,4,8,1-tetraazacyclotetradecane-N,N′,N″,N″-tetraacetic acid (TETA) has been used for assessment of ^64^Cu specific activity as it binds copper 1:1; the same method can be adopted for ^67^Cu analysis [5,65]. The effective specific activity of ^64^Cu was assessed based on the percent complexation of ^64^Cu-TETA as a function of TETA concentration, monitored by radio-TLC. Other metallic impurities can’t be determined in this way; therefore, Ohya et al. [50] developed the so-called post-column analysis for the evaluation of ^67^Cu chemical purity, based on ion chromatography in combination with an ion-pair reagent and UV detector.

## 3. The Use of ^67^Cu for Medical Applications

### 3.1. Chelators for Copper

The coordination chemistry of the transition metal copper in an aqueous solution is limited to three accessible oxidation states (I–III). The lowest oxidation state, Cu(I) allows formation of complexes with various ligands, which are however labile and lack sufficient stability for radiopharmaceutical applications. Cu(II) is less labile toward ligand exchange and is optimal for incorporation into radiopharmaceuticals. The Cu(III) oxidation state is relatively rare and difficult to attain and thus not of importance for radiopharmaceuticals [71,72]. To evaluate ^67^Cu for radioimmunotherapy (RIT), the macrocyclic chelating agent 1,4,7,11-tetraazacyclotetradecane-*N*,*N*′,*N*″,*N**‴*-tetraacetic acid (TETA) was designed specifically to bind copper rapidly and selectively for conjugation to MoAbs. The synthesis of this “bifunctional” metal chelator, 6(*p*-bromoacetamidobenzyl)-1,4,8,11-tetraazacyclotetradecane-*N*,*N*′,*N*″,*N*‴-tetraacetic acid, which can be covalently attached to proteins and which binds copper stably in human serum under physiological conditions, was reported by Moi et al., in 1985 [73]. In contrast, other chelators based on ethylenediaminetetraacetic acid or diethylenetriaminepentaacetic acid rapidly lose copper to serum albumin under the same conditions.

Despite their use in several pre-clinical studies, the stability of Cu(II) metal complexes with common macrocyclic ligands, such as DOTA (1,4,7,10-tetraazacyclododecane-1,4,7,10-tetraacetic acid) and TETA was evaluated and considered not satisfactory for biomedical applications, demonstrating that nearly 70% of the ^64^Cu trans-chelated in the liver 20 hours post injection [74,75,76]. 

The addition of a bond (or Cross-Bridge, CB) between two opposite nitrogen atoms in the tetraazamacrocyclic skeleton of the macrocycle gives greater stability. The Cu(II) complex with the ligand CB-TE2A (4,11-bis(carboxymethyl)-1,4,8,11-tetraazabicyclo [6.6.2] hexadecane) has been demonstrated to be particularly stable in vivo and to have exceptional inertness kinetics in aqueous solution thanks to the stabilizing effects given by the CB-macrocycle and by the two hanging carboxymethyl arms [77,78,79,80]. Unfortunately, the extreme labeling conditions at 95 °C do not make it a good candidate for the development of radiopharmaceuticals labile at high temperatures, e.g., antibodies [81]. The replacement of one or both of the hanging carboxymethyl arms with groups derived from phosphonic acid (–CH_2_-PO_3_H_2_), CB-TE1A1P (4,8,11-tetraazacyclotetradecane-1-(methanophosphonic acid)-8-(methanocarboxylic acid)), and CB-TE2P (1,4,8,11-tetraazacyclotetradecano-1,8-bis(methanephosphonic acid)), has been shown to confer greater thermodynamic and kinetic stability, as well as greater selectivity and faster complexation kinetics [82]. 

Another class of potential sarcophagus-like cage structure chelators for copper are hexaazamacrobicyclic ligands [83,84]. Labeling conditions of copper complexes with sarcophagine-like ligands are particularly favorable in a wide pH range (4–9) with yields of 100% in a few minutes at room temperature [75]. The complexes obtained possess excellent in vitro and in vivo stability [85,86,87]. 

Some N2S2-type acyclic chelators have also been studied, such as PTSM (pyruvaldehyde-bis(N4-methylthiosemicarbazone)) and ATSM (diacetyl-bis(N4-methylthiosemicarbazone)). This class of chelators, bis (thiosemicarbazones), act as tetradentate ligands and coordinate Cu^2+^ forming stable and neutral complexes, with a square planar geometry, which show a high permeability of the cell membrane. The preparation of these complexes with radioactive copper isotopes can be carried out under mild conditions in high yield [74,87,88,89]. 

### 3.2. Pre-Clinical Studies

In the early studies on RIT, the ^67^Cu labeled radioimmunoconjugate, ^67^Cu-2IT-BAT-Lym-1, was prepared by conjugating the bifunctional TETA derivative BAT to antibody Lym-1, monoclonal antibody against human B cell lymphoma, via 2-iminothiolane (2IT). This modification did not significantly alter immunoreactivity of the antibody [90]. Nude mice bearing human Burkitt’s lymphoma (Raji) xenografts treated with ^67^Cu-2IT-BAT-Lym-1 achieved high rates of response and cure with modest toxicity [91]. In the pre-clinical radioimmunotherapy comparison with another antibody conjugate, BAT-2IT-IA3, the therapeutic effect of ^67^Cu was similar to that of ^64^Cu [92]. 

Searching for optimal radiocopper labeling of anti-L1-CAM antibody chCE7, five bifunctional copper chelators were synthesized and characterized (CPTA-*N*-hydroxysuccinimide, DO3A-l-p-isothiocyanato-phenylalanine, DOTA-PA-l-p-isocyanato-phenylalanine, DOTA-glycyl-l-p-isocyanato-phenylalanine and DOTA-triglycyl-l-p-isocyanato-phenylalanine). CPTA-labeled antibody achieved the best tumor to normal tissue ratios when biodistributions of the different ^67^Cu-chCE7 conjugates were assessed in tumor-bearing mice. High resolution PET imaging with ^64^Cu-CPTA-labeled MAb chCE7 showed uptake in lymph nodes and heterogeneous distribution in tumor xenografts [93]. The therapeutic value of ^67^Cu was also demonstrated pre-clinically in the treatment of bladder cancer [94]. It has been demonstrated that the chelator makes the difference in the behavior of conjugates [95]; therefore, new chelator/biological vector compositions are under investigation. 

In recent years, new inputs came from the use of the cage ligand, sarcophagine-like (Sar), as a chelator for ^67^Cu. The studies on ^67^Cu-CuSarTATE, a somatostatin receptor targeting ligand and sarcophagine-containing PSMA ligand, labeled either with ^64^Cu- or ^67^Cu, supported translation of these radiopharmaceuticals to the clinics [96,97,98,99].

### 3.3. Clinical Studies

As early as in 1972, the absorption of copper was determined by the simultaneous administration of ^64^Cu orally and ^67^Cu intravenously to patients with Wilson’s disease. The disease results in excess accumulation of copper in tissues, such as the liver and brain. However, in this study, copper radionuclides were administered in a simple cationic form [100]. Two decades later, following the promising pre-clinical results, the radioimmunoconjugate ^67^Cu-2IT-BAT-Lym-1 was introduced to the clinics [101]. Then the phase I/II clinical trial of ^67^Cu-2IT-BAT-Lym-1 was conducted, in an effort to further improve the therapeutic index of Lym-1-based radioimmunotherapy patients with B-cell non-Hodgkin’s lymphoma (NHL). ^67^Cu-2IT-BAT-Lym-1 provided good imaging of NHL and favorable radiation dosimetry. The mean radiation ratios of tumor to body and tumor to marrow were 28:1 and 15:1, respectively. Tumor-to-lung, -kidney, and -liver radiation dose ratios were 7.4:1, 5.3:1, and 2.6:1, respectively. This ^67^Cu-2IT-BAT-Lym-1 trial for patients with chemotherapy-resistant NHL had a response rate of 58% (7/12). No significant nonhematologic toxicity was observed. Hematologic toxicity, especially thrombocytopenia, was dose limiting [102]. Recently, there have been several new completed and ongoing clinical trials with ^67^Cu radiolabeled compounds. These included the theranostic trial for pre-treatment dosimetry of somatostatin analog ^64/67^Cu-SarTATE [103] and peptide receptor radionuclide therapy using ^67^Cu-SarTATE in pediatric patients [104,105].

The decay properties of ^67^Cu were also explored in order to improve the patient management in the hospital, going into two critical steps further: demonstrating the therapeutic efficacy of ^67^Cu-PRIT (pre-targeted radioimmunotherapy) and illustrating the usefulness of pre-targeted ^64^Cu-PET as a predictive indicator of response to ^67^Cu-PRIT [106]. The use of γ-emissions of ^67^Cu at 184.6 keV (48.7%) and rarely 93.3 keV (16.1%) for SPECT imaging, assuming that the branching ratios of ^67^Cu’s γ-emissions are much higher than those of ^177^Lu, is expected to provide higher imaging sensitivity [107,108]. It is relevant to underline the recent work on SPECT imaging with Derenzo phantom, to study SPECT image quality using ^67^Cu [10]. The use of a medium energy (ME) collimator is typically recommended for ^177^Lu, since its γ-ray has an energy of 208 keV, while for the 140 keV γ-ray emitted by ^99m^Tc, the gold-standard radionuclide for SPECT imaging, the low-energy high-resolution (LEHR) collimator is recommended. As the γ-emission energy of ^67^Cu (185 keV, Table 1) falls between these two aforementioned energies, the appropriate collimator for SPECT imaging with ^67^Cu was found to be the ME collimator. Although there is a reduced image quality for ^67^Cu and ^177^Lu, these radionuclides are still considered adequate for tumor identification, suggesting that the post-treatment dosimetry is possible for ^67^Cu-labeled radiopharmaceuticals as it is for ^177^Lu [10]. 

## 4. Discussion

Comparing with the accelerator-produced mode, the routine production of ^67^Cu via the (n,p) reaction in a nuclear reactor is less profitable due to the relatively low yields and expensive ^67^Zn target material. Only a limited number of nuclear reactors offer the fast neutron flux higher than 10^14^ n cm^−2^s^−1^ and only a few higher than 10^15^ n cm^−2^s^−1^, which would be preferred for the more efficient production of ^67^Cu [40]. For this reason, the reactor route is currently not expected to contribute significantly to the availability of ^67^Cu, though reactor produced ^67^Cu might be useful as a tracer in research phase on the development of new radiopharmaceuticals or to expand research globally [43]. In contrast, the number of LINACS and cyclotrons with high energy protons is increasing [109,110], thus rapidly filling the ^67^Cu availability gap. It is worth to mention that the scarce availability of intense deuteron-beams with energy higher than 9 MeV is curtailing the use of the ^70^Zn(d,x)^67^Cu reaction. A more thorough detailed insight into the economy of charged-particle induced reactions is given in Table 5, where the target material costs were estimated considering the target thicknesses, a hypothetical beam spot area of 1 cm^2^ and an average price of enriched ^68^Zn about USD 3/mg, ^70^Zn about USD 13/mg and ^64^Ni about USD 30/mg. Table 5 also reports the estimated cost of ^67^Cu activity for each case. However, these estimates did not include the influence of recovery and reuse of the irradiated enriched target material. Obviously, the ^67^Cu cost will decrease if the same enriched material will be re-used for several production cycles. The number of these cycles has to be carefully studied, to assure a final ^67^Cu product accomplishing the regulatory requirements.

The calculations revealed that when the target material is fully enriched in the desired isotope (i.e., 100% enrichment, unless for the ^64^Ni case that is reported from [24] with a 98% enrichment), the convenient route to obtain ^67^Cu (without ^64^Cu coproduction) is by using deuteron beams and ^70^Zn targets. On the other hand, if some ^64^Cu coproduction is acceptable for clinical applications, the ^68^Zn(p,2p)^67^Cu reaction seems to be a better option, since it provides a larger ^67^Cu yield at a lower price per GBq (mCi) in comparison with the ^70^Zn(d,x)^67^Cu route. The proton-induced reaction on a multi-layer target, composed of ^68^Zn and ^70^Zn, maximizes the ^67^Cu yield at a reasonable price per GBq (or per mCi); however, there is a coproduction of ^64^Cu. This route has been previously inhibited by the lack of commercial availability of electron accelerators and the need for an enriched thick target that must be recycled. Electron accelerators are now becoming commercially available and methods for target recycling have been worked out which make this route of production more cost effective. It is important to note that the dose of ^67^Cu radioactivity required for one treatment is about 3.7 GBq (100 mCi) [43], but it can vary depending on the specific case and radiopharmaceutical. The ^64^Cu coproduction and its impact on the dose delivery has to be carefully considered for each radiopharmaceutical, taking into account the specific biodistribution and timing for wash-out [111]. In general, the limit for the RNP is 99% and the dose increase due to contaminant radionuclides is 10%; however, considering the ^64^Cu and ^67^Cu decay characteristics (i.e., the emission of β^-^ particles and Auger electrons), preclinical studies with ^64/67^Cu-radiopharmaceuticals are encouraged to determine the potential impact of this combined therapy.

It is worth noting that pure ^67^Cu is available using the photo-induced production route [10]. Its major drawback is the need of a large, massive enriched ^68^Zn target, having not only an economic impact on the initial investment but also a technological one on the radiochemical processing and target recovery. In addition to this route, it is important to mention the possibility of using an online mass separator to select ^67^Cu, to later apply the chemical separation of Cu-isotopes from the collected ^67^X radionuclides. 

Mass separation, in contrast to the commonly used ion exchange and extraction chromatography, is expected to increase the availability of certain “exotic” radionuclides, among them ^67^Cu. This novel approach will be studied within the recently granted EU project (PRISMAP) [112]; however, the efficacy of this process is yet to be shown.

## 5. Conclusions

This review reveals the international effort to supply ^67^Cu, a promising theranostic radionuclide. The increasing availability of intense particle accelerators and the optimization of the associated technologies (targetry and radiochemical processing) are making ^67^Cu closer to the clinics. The positron emitter counter-parts of ^67^Cu, i.e., ^60^Cu, ^61^Cu, and ^64^Cu, can be produced in cyclotrons. In particular, ^64^Cu is now widely available for clinical use, promoting the development of innovative Cu-labeled radiopharmaceuticals. The improved availability of ^67^Cu would speed up further radiopharmaceutical applications for therapy. Should ^67^Cu be produced in sufficient quantities and quality, its clinical use would spread worldwide. Therefore, in addition to a detailed analysis of the possible nuclear reactions to produce ^67^Cu, the radiochemical procedures to extract and purify Cu from the bulk material were also described in this work. Recent developments in the photoproduction of ^67^Cu, and in the possibility of having accelerators providing intense 70 MeV proton beams and/or intense 30 MeV deuteron beams, are grounds for a future reliable supply of ^67^Cu. 

In most of the nuclear reactions leading to ^67^Cu, the ^64^Cu is produced in various radioactivity ratios. What is the impact of this impurity on a patient’s dosimetry? Further studies on the possible use of ^67/64^Cu labeled-radiopharmaceuticals are encouraged, to find out the possible therapeutic benefits for the patient exploiting, at the same time, the β^−^ radiation emitted by ^67^Cu and ^64^Cu decay and the shorter range Auger-electrons emitted by ^64^Cu. Encouraging technological achievements, namely working out methods to isolate the small amount of ^67^Cu from the large mass of zinc in the target and to recover the target material in the accelerator-based production, show that, soon, ^67^Cu will be available daily in the United States and in Europe for research purposes. The relatively long half-life of ^67^Cu makes production in large facilities and the shipping of the purified product to clinical centers possible. 

## Figures and Tables

**Figure 1 molecules-27-01501-f001:**
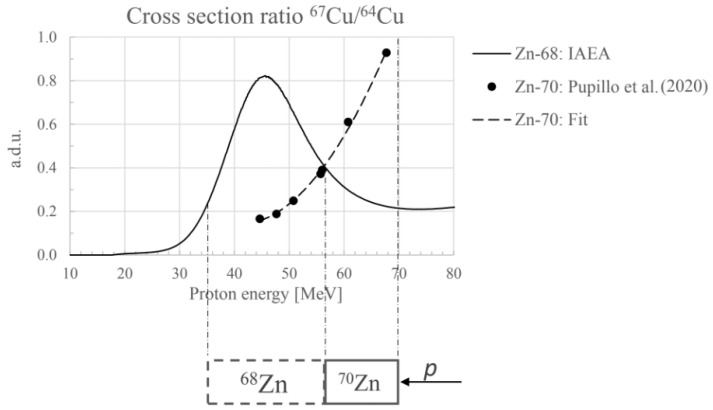
Plot of the nuclear cross section ratio for the production of ^67^Cu and ^64^Cu radionuclides: the continuous line is the IAEA recommended value for ^68^Zn targets; the dashed line refers to the measured values for ^70^Zn targets. The vertical dashed lines refer to the favorable energy range for ^67^Cu production. A scheme of the multi-layer target configuration described in the international INFN patent is shown at the bottom [21].

**Table 1 molecules-27-01501-t001:** Main decay characteristics of ^67^Cu and ^64/61/60^Cu-radionuclides [7].

	Half-Life	Main γ-ray Energy, Intensity (keV) (%)	Mean β^+^ Energy, Intensity (keV) (%)	Mean β^−^ Energy, Intensity (keV) (%)	Auger and IC Electrons
^67^Cu	61.83 h	184.577 (48.7)	-	141 (100)	Yes
^64^Cu	12.701 h	1345.77 (0.475)	278 (17.6)	191 (38.5)	Yes
^61^Cu	3.336 h	282.956 (12.7) 656.008 (10.4)	500 (61)	-	Yes
^60^Cu	23.7 m	826.4 (21.7) 1332.5 (88.0) 1791.6 (45.4)	970 (93)	-	Yes

**Table 2 molecules-27-01501-t002:** ^67^Cu and ^64^Cu production yields obtained using proton-beams on ^70^Zn and ^68^Zn enriched materials for each target material individually and the multi-layer target configuration (I = 1 µA; T_IRR_ = 62 h).

Target	Energy Range (MeV)	^67^Cu @ EOB (MBq/µA)	^64^Cu @ EOB (MBq/µA)	^67^Cu/(^64^Cu + ^67^Cu) @ EOB	^67^Cu/(^64^Cu + ^67^Cu) @ 24 h Post EOB
^70^Zn	25–10 [9]	2.13 × 10^2^	-	100%	100%
^68^Zn	70–35 [9]	1.24 × 10^3^	6.51 × 10^3^	16%	35%
^70^Zn + ^68^Zn	70–55 + 55–35 [16]	1.86 × 10^3^	5.71 × 10^3^	25%	48%

**Table 3 molecules-27-01501-t003:** ^67^Cu and ^64^Cu production yields obtained by using proton-, deuteron-, and α-beams on ^70^Zn, ^68^Zn, and ^64^Ni enriched target materials assuming I = 30 µA and T_IRR_ = 24 h.

Beam	Target	Energy Range (MeV)	Thickness (mm)	^67^Cu @ EOB (MBq)	^64^Cu @ EOB (MBq)
Protons	^70^Zn	25–10	1.22	3.01 × 10^3^	-
	^68^Zn	70–35	6.43	1.75 × 10^4^	1.48 × 10^5^
	^70^Zn + ^68^Zn	70–55 + 55–35	3.26 + 3.27	2.62 × 10^4^	1.30 × 10^5^
Deuterons	^70^Zn	26–16	0.58	4.01 × 10^3^	-
Alpha	^64^Ni	30–10	0.16	1.00 × 10^3^	-

**Table 4 molecules-27-01501-t004:** Cu/Zn separation and purification procedures (SE = solvent extraction, IE = ion exchange; TTA = thenoyltrifluoroacetone), processing time and process yield are included, if available.

Ref.	Target	Dissolution	Radiochemical Separation Method	Processing Time	Yield
[36,53]	^nat^Zn foil or ZnO	conc. HCl	SE with dithizone	-	>90%
[56]	^nat^Zn plates	30% HCl (400 K)	SE with dithizone +IE with AG 50 W +IE with AG1-X8	5 h	85 ± 20% for SE
[57]	ZnO (28–30 g)	conc. HCl	SE with dithizone +IE with AG1-X8	5–7 h	>90%
[44]	^67^ZnO	1 N HCl +30% H_2_O_2_	SE with TTA	-	-
[51,54]	^68^ZnO (100 mg)	conc. HCl	IE with Dowex 1 × 8	4 h	94%
[52]	^nat^Zn (1–2 g)	8 M HCl	IE with AG1-X8	2 h	95%
[42]	^67^ZnO, 93.4% (50 mg)	4 M HCl	IE with Diaion SA-100	-	95%
[13]	^68^Zn, 99.7% (0.7–4 g) electroplated on Ti or Al	12 M HCl	IE with AG50-X4 +Chelex-100 +AG1-X8	-	92–95%
[56]	^nat^Zn plate	37% HCl	IE with AG50W +Chelex-100 +AG1-X8	4.5 h	90%
[16]	^70^Zn, >95% metal foils	10 M HCl	IE with AG50W-X4 +AG1-X8	4 h	95 ± 2%
[63]	^nat^ZnO powder (3.5 g)	10 M HCl (100 °C)	Double coprecipitation with AgNO_3_	<3 h	81 ± 6%
[31]	^68^Zn metal ingot target (100 g)	-	Sublimation	Rate of Zn separation from Cu: >50 g/h	Removing of >99% of Cu (and other metals) from Zn in each sublimation cycle
[56]	^nat^Zn plate	30% HCl (400 K)	Electrolysis	12 h	60%
[2]	^nat^Zn foil	conc. HCl +HNO_3_	Electrolysis + IE with MP-1	-	80%
[39]	^67^ZnO, ≥94% (50–100 mg)	1 M H_2_SO_4_	Spontaneous electrochemical separation	1.5 h	95%

**Table 5 molecules-27-01501-t005:** ^67^Cu activity cost ($ (USD)/GBq) by using proton-, deuteron-, and alpha-beams on ^70^Zn, ^68^Zn, and ^64^Ni enriched materials, considering I = 30 µA, T_IRR_ = 24 h and enriched targets.

Beam	Target	Energy Range (MeV)	Thickness (mm)	Target Cost ($)	^67^Cu @ EOB (GBq) (mCi)	^67^Cu Cost ($/GBq) ($/mCi)	^64^Cu
Protons	^70^Zn	25–10	1.22	11,284	3 (81)	3761 (139)	-
	^68^Zn	70–35	6.43	13,758	17.5 (473)	786 (29)	Yes
	^70^Zn + ^68^Zn	70–55 + 55–35	3.26 + 3.27	30,186 + 7000	26.2 (709)	1420 (52)	Yes
Deuterons	^70^Zn	26–16	0.58	6500	4.1 (110)	1323 (49)	-
Alpha	^64^Ni	30–0	0.16	3300	1 (27)	4272 (158)	-

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
