# Peer review of "67Cu Production Capabilities: A Mini Review"

_molecules, 2022, doi:10.3390/molecules27051501_

Round 1

Reviewer 1 Report

This review to my view is very well done and written very clearly way. It stablish an actual and informed state of the art, assembling the many research efforts on field. The work was prepared with criteria and is very well presented. It could be published in the present form but I have 4 smalls comments that could be considered.

Comments

  • Is not clear for me why ranges are not expressed in the same format; same times in format lower limit – higher limit and other in the opposite order, sometimes for direct comparison as  in table 2;
  • In line 193, page 6, we read that “71Ga targets are unusual for 67Cu production”. That is strictly correct but according to argument used by the authors there is a strong and unescapable reason for that. So the statement could be rephrased because 71Ga targets more than unusual for 67Cu production seems to be unviable or impracticable.
  • In line 447, page 15, appears the  acronym RIT (radioimmunotherapy) defined in the introduction (page 2, line 38) and used thereafter; the authors should valuate   to do the definition of the acronym just in page 15;
  • Table 4 (form page 9, line 328 to page 12, line 330). In this format, the table is almost unreadable and the valuable information contained is of difficult comparison. The authors should consider a different solution perhaps to delete the column with the protocol and to add an appendix with that information.

Typing correction

  • In line 170, page 5, appears “omixture” that should be correct to “misture”;

Author Response

We would like to thank the reviewer for the valuable comments. The point by point response letter is included. 

Reviewer 2 Report

This paper reviews the current state of Cu67 production routes from the target materials and nuclear reactions to separation and purification. Recent Cu67 radiopharmaceutical production is also described. This is an important and growing area of research with commercial and clinical implications. The authors don't provide much opinion about the long term and widespread feasibility of this radionuclide, perhaps because there are still too many unknowns. The paper is overall well-organized with up to date references. There are some typographical and grammatical errors that will need to addressed. Below are some comments:

  1. Despite there being a discussion towards the end of the manuscript, a short discussion or summary would be useful at the end of each section. For example, provide a discussion of the values in Table 3 at the end of section 2.1.1.
  2. Table 4. Where possible, add a column or statement outlining the duration for each procedure. And a summary and conclusions for section 2.3.2.
  3. Line 370. Include a definition for I gamma. Does 48% refer to I gamma?
  4. Line 382, This paragraph has a few typographical and grammatical errors. For example, ‘choosen’, ‘cooper’ ‘being copper ubiquitous’.
  5. Line 409. Change remind to remember
  6. Line 417. Include full name of DOTA here.
  7. Line 474. Should that be ‘faster’ complexation kinetics?
  8. Line 510. The cage ligand should be referred to as sarcophagine (Sar), not SarAr. SarAr is a particular Sar derivative.
  9. Line 513. How did the authors reach this conclusion? More discussion of this study is needed.
  10. Line 520. The statement ‘in liver disorder in copper excretion’ needs to be rewritten
  11. Line 513. It should be noted that 67Cu-octreotate is the same compound as 67Cu-SarTATE.
  12. Line 569. My feeling is that it should be written $3/mg
  13. Line 596. Has there been a cost analysis on the photo-induced production? The authors mention that commercial entities are investigating this route but there isn’t much discussion about why that is or how feasible this method is for widespread availability of Cu67.
  14. Line 625. Mention of the beta minus emission of Cu64 should be made. PNAS, 2001, 98, 3, 1206.
  15. Line 626. What technological achievements are being referred to here?
  16. References 123-125 could not be found in the text.

Author Response

(The authors gave the same response as above.)

Reviewer 3 Report

The presented review paper addresses an important issue related to the production strategies of a promising medicinal radionuclide of Copper, namely Cu-67, which can open up additional tools for radiotherapy and diagnostics. The paper is competently written and I believe that it will be very useful for many radiochemists and specialists in nuclear medicine.

I would suggest minor revision related with the added subsections 2.4. Chelators for copper; 2.5. Pre-clinical studies and 2.6. Clinical studies, all under the section 2. Production methods of 67Cu . This doesn't look appropriate and is a little misleading. Also, neither in the abstract nor in the introduction any hint is given that the minireview will also address these topics. To correct this inconsistency, I would suggest two options: to add a separate section on Medical uses of Cu-67 radiopharmaceuticals (and properly add this in the abstract and in the introduction), or leave their minireview as intended and strictly focused on the production processes and provide extensive citations on most recent review papers summarizing the recent achievements in development of Chelators for copper; Pre-clinical studies and Clinical studies. To me the second option is more appropriate, taking into account the amount of review papers on these topics. Thereby, this minireview will be one of the few that focus specifically on the production methods, which is the limiting step in the further expansion of the medical use of Cu-67.

Herein, I attach the file with highlighted parts, which need some attention from the authors - mainly to correct misspelling or inconsistency in Br or Am English, add some missing abbreviations, etc.. 

In conclusion, I think this is a useful minireview and should be published after considering the suggested revision.

Author Response

(The authors gave the same response as above.)
